# Metabolomics Approach on Non-Targeted Screening of 50 PPCPs in Lettuce and Maize

**DOI:** 10.3390/molecules27154711

**Published:** 2022-07-23

**Authors:** Weifeng Xue, Chunguang Yang, Mengyao Liu, Xiaomei Lin, Mei Wang, Xiaowen Wang

**Affiliations:** Technical Center of Dalian Customs, Dalian 116000, China; 2004ycg51@163.com (C.Y.); laoyao1024@163.com (M.L.); lynn9857@163.com (X.L.); monkeywangcn@sina.com (M.W.); wxw0652@sina.com (X.W.)

**Keywords:** metabolomics, marker compounds, non-targeted screening, pharmaceutical and personal care products, plant-derived food

## Abstract

The metabolomics approach has proved to be promising in achieving non-targeted screening for those unknown and unexpected (U&U) contaminants in foods, but data analysis is often the bottleneck of the approach. In this study, a novel metabolomics analytical method via seeking marker compounds in 50 pharmaceutical and personal care products (PPCPs) as U&U contaminants spiked into lettuce and maize matrices was developed, based on ultrahigh-performance liquid chromatography-tandem mass spectrometer (UHPLC-MS/MS) output results. Three concentration groups (20, 50 and 100 ng mL^−1^) to simulate the control and experimental groups applied in the traditional metabolomics analysis were designed to discover marker compounds, for which multivariate and univariate analysis were adopted. In multivariate analysis, each concentration group showed obvious separation from other two groups in principal component analysis (PCA) and orthogonal partial least squares discriminant analysis (OPLS-DA) plots, providing the possibility to discern marker compounds among groups. Parameters including S-plot, permutation test and variable importance in projection (VIP) in OPLS-DA were used for screening and identification of marker compounds, which further underwent pairwise *t*-test and fold change judgement for univariate analysis. The results indicate that marker compounds on behalf of 50 PPCPs were all discovered in two plant matrices, proving the excellent practicability of the metabolomics approach on non-targeted screening of various U&U PPCPs in plant-derived foods. The limits of detection (LODs) for 50 PPCPs were calculated to be 0.4~2.0 µg kg^−1^ and 0.3~2.1 µg kg^−1^ in lettuce and maize matrices, respectively.

## 1. Introduction

Pharmaceutical and personal care product (PPCP) contamination in animal-derived foods has attracted worldwide attention, and a series of formal regulatory documents on the maximum residue limits (MRLs) of PPCPs from different countries and organizations has been issued [1,2,3,4]. However, PPCPs-induced contamination in plant-derived foods has not been fully addressed [5]. Previous studies [6,7,8,9,10,11,12,13,14] indicate that some plant-derived foods (e.g., corn, barley, pea, wheat, carrot, potato, cucumber and lettuce) can easily absorb PPCPs from soil with animal manure used as a fertilizer, which contains several kinds of commonly used antibiotics, e.g., tetracyclines, quinolones, sulfonamides and β-lactam, with their total concentration from the µg kg^−1^ to the mg kg^−1^ level in the plants [9,15,16,17,18]. Due to the lack of evaluation standards of PPCPs in plant-derived foods, it is hard to directly judge whether the residue concentrations of PPCPs can induce adverse effects on human health. Referring to the regulatory files on MRLs of PPCPs in animal-derived foods [2,4], which proposed a concentration of 10 µg kg^−1^ as the threshold of safety for most PPCPs, it can be inferred that if the concentrations of PPCPs in plant-derived foods exceed 10 µg kg^−1^, it triggers a food safety risk. Therefore, the top priority is to develop reliable analytical methods for the investigation of PPCP residues in plant-derived foods.

Previous studies [6,9,10,19,20] have proposed some analytical methods based on high-performance liquid chromatography-tandem mass spectrometer (HPLC-MS/MS) for PPCPs detection in plant matrices. These studies mainly focused on the contamination of antibiotics, especially for tetracyclines, quinolones and sulfonamides. Most methods are customized and showcase the excellent detection performance for specific PPCPs, but are invalid for detecting other PPCPs not in the customized database. With the rapid development of the modern chemical industry’s ability to synthesize new compounds, there is reason to believe that more and more PPCPs will be produced and applied in animal husbandry; as a result, continuous uptake of PPCPs by plant-derived foods will probably lead to more complicated, serious and underlying food safety risk. The United States, China and Japan, as the world’s top three economies, plus the European Union, have issued regulatory documents on MRLs of only 95, 128, 180 and 139 PPCPs in animal-derived foods, and some listed PPCPs are of repeated emergence [1,2,3,4]. The sticking point is that the existing technologies cannot meet the detection requirements for increasing unknown and unexpected (U&U) PPCPs, for which the most effective method is to develop non-targeted screening methods, as proposed by the NORMAN network (www.norman-network.net, accessed on 20 November 2021) founded in 2005 by the European Commission [21,22].

Non-targeted screening can be defined from the narrow and broad senses. The former is reliant on the established screening database to discern contaminants [23]. The contaminants in the database are known, but those existing in the matrix are obscure, thus the screening practicability depends on the database size. The latter sense is to employ omics-related approaches to complete U&U contaminant screening [24,25], which can be realized by high resolution mass spectrometry (HRMS) technology [26,27]. To date, LC-MS/MS-based metabolomics analytical methods have showed good practicability on non-targeted screening of some pesticides in food matrices, e.g., orange juice [28], milk [24] and tea [29], with the screening ratio of pesticides depending on their contents. These studies have obtained desirable outcomes, but the methods they proposed are so sophisticated that they are not favorable for wide application. Nowadays, the development of non-targeted metabolomics analysis still encounters many great obstacles, especially for data analysis, which is the bottleneck to be urgently solved through the advancement of data processing tools and improvement of HRMS data quality.

In view of this, we developed a novel metabolomics-based analytical method via seeking marker compounds on behalf of 50 PPCPs as U&U contaminants spiked in lettuce and maize matrices to achieve non-targeted screening. Ultrahigh-performance liquid chromatography-tandem mass spectrometer (UHPLC-MS/MS) was used to obtain output results for metabolomics analysis. Herein, 14 sulfonamides, 12 quinolones, 10 nitroimidazoles, 7 agonists, 4 steroids and 3 tetracyclines were selected as target PPCPs, in which quinolones, sulfonamides and tetracyclines are of relatively high detection frequency in plant-derived foods [9,15,16,17,18]. Lettuce and maize are consumed in high quantities worldwide, and have proved to easily absorb PPCPs from the soil [6,19]. Lack of formal documents to regulate the MRLs of PPCPs in plant-derived foods makes it difficult to directly evaluate whether the contents of PPCPs in the foods are in the safety range. According to the guidelines of GB 31650-2019 [4] and Commission Regulation (EU) No 37/2010 [2], the MRLs of most PPCPs in animal-derived foods are no lower than 10 µg kg^−1^, which was used as the test concentration of 50 PPCPs in our study to perform screening analysis. The goal of this study is to develop an applicable analytical method on the basis of metabolomics, which can accurately, rapidly and comprehensively achieve the screening and identification of potential non-targeted contaminants in plant-derived foods.

## 2. Materials and Methods

### 2.1. Chemicals and Materials

The lettuce was bought from a local market in Dalian City. Ethylenediamine tetraacetic acid disodium salt (Na_2_EDTA), citric acid, sodium hydrogen phosphate (Na_2_HPO_4_), anhydrous sodium sulfate (Na_2_SO_4_), sodium chloride (NaCl), sodium hydroxide (NaOH), hydrochloric acid (HCl) and C18 powder (Sinopharm Chemical Reagent Co., Ltd., Shanghai, China); methanol and acetonitrile (HPLC grade, Merck, Darmstadt, Germany); formic acid (HPLC grade, Shanghai ANPEL Laboratory Technologies Inc., Shanghai, China); filter membrane (0.22 µm, Agilent Technologies, Singapore, MI, USA); ultrapure water (Milli-Q ultrapure water system, Merck, Darmstadt, Germany); ciprofloxacin-d8 hydrochloride solution (100 µg mL^−1^ in methanol, First Standard, Ridgewood, NY, USA). Analytical standard compounds for 50 PPCPs (purity > 98.3%) were obtained from First Standard (Ridgewood, NY, USA), Sigma (Alexandria, VA, USA), TRC (Toronto, ON, Canada) and Dr. Ehrenstorfer (Augsburg, Germany). More details on the 50 PPCPs are shown in Table 1.

### 2.2. Solution Preparation

A total of 50 PPCPs were separately prepared with methanol at 100 µg mL^−1^, 1 mL of which was withdrawn, mixed together and further diluted with methanol to obtain a 1 µg mL^−1^ solution. Then, 100 ng mL^−1^ ciprofloxacin-d8 methanol solution was prepared by diluting its 100 µg mL^−1^ solution. A 0.1 mol L^−1^ Na_2_EDTA-Mcllvaine buffer solution was prepared with Na_2_HPO_4_ (5.5 g), citric acid (12.9 g) and Na_2_EDTA (37.2 g) dissolved in 1 L pure water, which was further adjusted to pH 4.0 with 0.1 mol L^−1^ HCl or NaOH solution.

### 2.3. Sample Preparation and Pretreatment Process

(a) Lettuce sample was cut into small pieces, then ground into batter by tissue homogenizer; (b) 2.0, 5.0 and 10.0 g lettuce batters, together with one-to-one corresponding 20, 50 and 100 µL of 50 PPCPs mixed solutions (1 µg mL^−1^) were poured into 50 mL polypropylene centrifuge tubes. To calibrate the recovery during the sample pretreatment process, ciprofloxacin-d8 methanol solution (0.5 mL, 100 ng mL^−1^) as recovery internal standard was further added, as adopted in previous studies [30,31,32]; (c) 5 mL Na_2_EDTA-Mcllvaine buffer solution (0.1 mol L^−1^) was dumped into the tube, vortexed for 1 min, then 20 mL 1% (*V*/*V*) formic acid/acetonitrile solution was added further, stirring for 1 min. An extraction salt package (10.0 g Na_2_SO_4_ + 2.0 g NaCl) was added for stratification under salting out after the solution standing for 10 min, centrifuging at 4500 r min^−1^ for 5 min; (d) then, after transferring all the supernatant into new 50 mL polypropylene centrifuge tubes, adding 100 mg C18 powder, vortexing for 1 min, centrifuging at 4500 r min^−1^ for 3 min, the solution was extracted to another 50 mL centrifuge tube, dried with N_2_ blowing by nitrogen blowing apparatus (N-EVAP-112, Organomation, Berlin, MA, USA), and redissolved in 1 mL 40% (*V*/*V*) methanol 0.1% formic acid/water solution, vortexed for 1 min; (e) then, filtered with a 0.22 µm filter membrane, the sample solutions of 50 PPCPs at the theoretical concentrations of 20, 50 and 100 ng mL^−1^ were prepared. Each concentration experiment was repeated nine times.

### 2.4. Sample Grouping and Naming

Samples of 20 ng mL^−1^–1~20 ng mL^−1^–9, 50 ng mL^−1^–1~50 ng mL^−1^–9 and 100 ng mL^−1^–1~100 ng mL^−1^–9 were employed to label samples from three concentration groups. Each sample provided a 30 µL solution as a quality control (QC) sample [29,33,34], which experienced 3 injections before and after each concentration group. As a result, 12 samples marked as QC-1, QC-2, and QC-12 were obtained to evaluate the stability of LC-MS/MS.

### 2.5. Analytical Method

The 50 PPCPs and ciprofloxacin-d8 were analyzed on a quadrupole/electrostatic field orbitrap LC-MS/MS system (Q Exactive Plus, Thermo Fisher Scientific Inc., Waltham, MA, USA) under the positive mode of electrospray ion (ESI) source. Components in the sample solution underwent separation within an Accucore RP-MS column (100 × 2.1 mm, 2.6 µm particle diameter, Thermo Fisher Scientific Inc., Waltham, MA, USA), with injection volume of 10 µL. Next, 0.1% (*V*/*V*) formic acid/water and 0.1% (*V*/*V*) formic acid/methanol solutions were prepared as the mobile phase A and B, respectively, with flow rate of 0.3 mL min^−1^. In consideration of the matrix complexity of lettuce and maize, there may be some impurities not eluted from the LC-MS/MS system in a relatively short time (738 s for the last eluted target PPCP in this study) designed only for 50 PPCPs, leading to the potential disruption for the elution and analysis of the next sample. Therefore, a longer elution program was designed as follows: gradient started from 5% B, kept for 2 min, then increased to 30% B in 1 min, at a duration of 7 min, further increased to 90% B in 1 min, holding on 25 min, finally decreased to 5% B in 1 min, equilibrating for 16 min. The oven temperature was set at 40 °C. Other parameter settings were as follows: heating and capillary temperature 320 °C; lens and spray voltage 50 and 3200 V, respectively; auxiliary and sheath gas N_2_, with flow rate at 10 and 40 arb, respectively; scan mode: full-scan/data-dependent two-stage scanning; MS parameters: full-scan resolution 70,000, maximum dwell time 100 ms, AGC target 1 × 10^6^, *m*/*z* scan range 100~1000; MS/MS parameters: resolution 17,500, maximum dwell time 50 ms, AGC target 2 × 10^5^.

LC-MS/MS output results of 50 PPCPs and ciprofloxacin-d8 were analyzed by Trace Finder 3.3 software, with screening conditions as follows: (a) for primary parent ion, signal to noise ratio 5.0, response intensity threshold 10,000, and mass error 5 ppm; (b) for secondary fragment ions, minimum matching number of ion 1, response intensity threshold 10,000, and mass error 5 ppm. On the basis of the peak area of the primary parent ion, ciprofloxacin-d8 was quantified with standard curve for recovery calculation.

### 2.6. Metabolomics Data Processing

LC-MS/MS was operated in full scan mode with RAW-formatted files as the direct output, which underwent conversion to corresponding mzXML-formatted files via the ProteoWizard software [35]. These new files are adaptable to the upload to the Workflow4Metabolomics (W4M) platform (https://workflow4metabolomics.usegalaxy.fr/, accessed on 20 November 2021) for metabolomics analysis [36]. After peak detection, alignment and retention time calibration, plus data normalization, centralization, scaling and transformation performed on the W4M platform, the data matrix was obtained in the format of variable and sample named as abscissa and ordinate, respectively [36,37]. Variable contains a series of information, e.g., molecular weight and retention time, with every marker compound corresponding to its unique variable, that is to say, the process to pursue marker compounds is actually a process to pursue eligible variables. Multivariate statistical analysis including principal component analysis (PCA) [38,39,40] and orthogonal partial least squares discriminant analysis (OPLS-DA) [41,42] was performed in SIMCA 14.1 software [43] after importing the data matrix. A permutation test with 200 iterations was employed for over-fitting judgement of the OPLS-DA model [43,44]. Other parameters to screen marker compound candidates include the absolute value of variable confidence in the S-plot plot [45] and variable importance in projection (VIP) [43,44,46], with the threshold above 0.9 and 1, respectively. After this, eligible marker compound candidates from 20 and 100 ng mL^−1^ groups can both be obtained, and only overlapped candidates in two groups, representing their significantly low and high concentration in the corresponding 20 and 100 ng mL^−1^ groups, were further investigated by pairwise *t*-test [47,48,49] in SPSS Statistics V17.0 software and fold change judgement for the univariate analysis. Univariate analysis is simple, intuitive and easy to be understood. It was used to quickly investigate the differences of marker compound candidates in different groups. To more rapidly verify the identity of marker compounds on behalf of 50 PPCPs, we directly compared the precise molecular weight (<5 ppm in absolute value of error), retention time and the adduct structure of marker compounds with that of the authentic 50 PPCPs (Table 1).

## 3. Results and Discussion

### 3.1. Data Preprocessing

As indicated in Figure 1, although only part of the total ion chromatograms at the retention time of 0~900 s is shown, during which all 50 PPCPs were eluted, obvious differences in peak intensity have already been observed in three concentration groups, implying the possibility to seek marker compounds among groups. The principle for relative standard deviation of peak intensity above 30% was employed to filter out invalid variables in QC and three concentration groups [50], with a final 6512 × 39 data matrix obtained for further analysis.

### 3.2. Multivariate Analysis

#### 3.2.1. PCA Analysis

As Taguchi [51] pointed out, PCA can make a natural classification for sample groups and eliminate the extreme data without knowing their categories, thus PCA can be used in metabolomics to assess the data quality and to identify outliers [38,39,40]. As indicated in Figure 2, no extreme data and outliers were observed. Samples at the same concentration gathered together, indicating the good classification of groups. Obvious separation among three concentration groups indicates the existence of major discrepancies, further paving the way to seek marker compounds from different groups.

#### 3.2.2. OPLS-DA Analysis

Theoretically speaking, the peak intensities of variables ought to increase with their rising concentrations, i.e., 20 and 100 ng mL^−1^ groups should present the minimum and maximum peak intensities, respectively. However, the reality may be different, due to the discrepancies in sample recoveries. Previous studies [30,31,32] proposed deuterated antibiotics as recovery internal standards to correct losses of PPCPs during sample preparation. In consideration of this, ciprofloxacin-d8 (parent ion *m*/*z* 340.19132; fragment ions *m*/*z* 296.20156, 253.15933 and 239.14367; retention time 6.73 min) was employed here to eliminate the peak intensity errors of variables induced by disparate recoveries of PPCPs during the pretreatment process. As shown in Appendix A, the recoveries of ciprofloxacin-d8 were calculated to be 80.1~85.9%, 80.3~86.2% and 81.6~87.7% in the 20, 50 and 100 ng mL^−1^ groups, respectively, based on the ciprofloxacin-d8 standard curve solutions (100, 50, 25, 10 and 5 ng mL^−1^) prepared in blank lettuce extract solution. After this, the recoveries of ciprofloxacin-d8 were all calibrated to 100% by multiplying a corresponding calibration coefficient, with which the peak intensities of ciprofloxacin-d8 were also calibrated, together with peak intensities for all the variables.

As shown in Figure 3, we can observe the separation of two camps on the first principal component axis. One camp represents the specific concentration group (green part), and the other camp is on behalf of the remaining two groups (blue part), indicating the existence of variables with significant differences between the two camps. Each point in the S-plot plots (Figure 4) represents a variable, which keeps away from the origin along *X*- and *Y*-axis, implying more contribution and higher confidence level of the variable to the difference. Therefore, the points at the two ends of ‘S’ can be deemed the most differentiating components. In the S-plot analysis, absolute value of confidence > 0.9 has been proposed to screen variables as marker compound candidates [45], which at the significantly low and high concentration should be searched at the right and left ends of S-plot plots in Figure 4a,b, respectively.

SIMCA 14.1 software performed permutation tests with 200 iterations to investigate whether the OPLS-DA models underwent data over-fitting, for which R^2^Y and Q^2^ are two common parameters to describe the interpretation level of the model in the *Y*-axis direction and the prediction level of the model [52,53], respectively. If R^2^Y and Q^2^ are both close (or equal) to 1, the OPLS-DA models are not susceptible to over-fitting. As can be seen from Figure 5, R^2^Y and Q^2^ values were no less than 0.991, indicating the good reliability, predictability and no over-fitting for all OPLS-DA models. VIP > 1 principle continues to screen marker compounds. Eventually, marker compounds on behalf of 50 PPCPs were all screened out as shown in Table 2. Negligible concentrations (<0.1 ng mL^−1^) of 50 PPCPs in the blank lettuce extract solution were obtained by the metabolomics analysis, which eliminates the interference of inherent (rather than spiked) 50 PPCPs residues in lettuce matrix to seek marker compounds.

### 3.3. Univariate Analysis

After multivariate analysis, a pairwise *t*-test [47,48,49] was firstly employed to examine whether marker compounds from a specific concentration group presented significant differences in peak intensity with those from other two groups. Pairwise *t*-test, as a reliable statistical test method, was performed to calculate *p* values between the two concentration groups and the *p* < 0.05 observed in this study indeed showed the existence of significant differences among groups. Previous studies [29,54] also adopted fold change of concentration > 2 to discern variables with high contrast among groups as marker compounds. Herein, marker compounds on behalf of 50 PPCPs all presented fold change values above 2, supporting the validity of marker compounds obtained with our analytical strategy.

The limits of detection (LODs) for 50 PPCPs were also considered here. Firstly, a 2.0 g blank lettuce sample was used to prepare an extract solution (1 mL) after the same pretreatment mentioned above. Then, a 20 ng mL^−1^ PPCPs solution was obtained by diluting their mixed methanol solution (20 µL, 1 µg mL^−1^) with 1 mL blank lettuce extract solution. The experiments were repeated in septuplicate to obtain seven samples, which underwent the same metabolomics analysis to obtain the peak intensities of 50 PPCPs. For each PPCP, a 20 ng mL^−1^ concentration level was deemed to correspond to average values of seven samples in peak intensity; therefore, the concentration (unit: ng mL^−1^) of each PPCP in a sample was calculated by its own peak intensity × 20/average peak intensity for the standard deviation measurement of the seven samples. According to the method proposed by US Environmental Protection Agency [55], the LOD values for 50 PPCPs were calculated to be 0.4~2.0 µg kg^−1^, as shown in Table 2.

### 3.4. Method Applicability in Maize Matrix

Maize as the primary food crop in China has proved to easily absorb PPCPs from the soil [19]; therefore, it was selected as another plant matrix different from vegetables to investigate the applicability of the developed metabolomics-based screening method. Maize sample was purchased from the local market and turned into a powder by a grinder. Then, it underwent the same above-mentioned pretreatment process after 50 PPCPs spiked at 10 µg kg^−1^ as well. Ciprofloxacin-d8 methanol solution (0.5 mL, 100 ng mL^−1^) was added for recovery calibration, with the results shown in Appendix A. The same metabolomics analysis was performed as indicated in Appendix A. Marker compounds to represent 50 PPCPs were also discovered (Appendix A), proving the good applicability of the metabolomics analytical method to non-targeted screening of various PPCPs residues in different plant matrices. As can be seen from Appendix A, the LOD values for 50 PPCPs in maize matrix were calculated to be 0.3~2.1 µg kg^−1^.

### 3.5. Real Sample Test

We collected lettuce and maize samples from six administrative districts including Zhongshan, Xigang, Shahekou, Gaoxin, Ganjingzi and Jinpu affiliated to Dalian City, each district with two sampling points. A total of 12 fresh lettuce samples were purchased from the local farmer’s market and immediately delivered to the laboratory for testing. The above process was also applied to the maize samples. After pretreatment experiments and metabolomics analysis, only one lettuce sample from Jinpu District was found to contain enrofloxacin and its content was 17.4 µg kg^−1^. Other samples had no detection of PPCPs. Although the detection rate of PPCPs in all the samples is only 1/24, and seemingly only one district is vulnerable to PPCPs contamination, the results are enough to show that our proposed method is competent for the screening of PPCPs in plant-derived foods. These spot check results alert us to the fact that PPCP-induced safety risk of plant-derived foods is on the horizon.

Previous studies have successfully applied non-targeted screening methods on the basis of metabolomics to pesticide residues in plant matrices, e.g., orange juice [28] and tea [29], providing the feasibility to screen PPCPs residues in plant-derived foods. In light of the otherness of analytes, the reported methods may not be completely applied to our study. Herein, we firstly considered spiked contaminants to be marker compounds and then implemented a marker compound-seeking analytical strategy of metabolomics to finish the non-targeted screening of contaminants in plant-derived foods, which is the biggest difference from previous studies [24,28,29]. Despite only 50 PPCPs and two plant matrices considered here, the developed method still has wide applicability due to the representation of these PPCPs and universal consumption of lettuce and maize.

Extensive use of PPCPs in livestock farming raises the risk that these compounds end up in soil where animal waste is used as fertilizer [9,56], which leads to the uptake of PPCPs by plant-derived foods from the soil [57,58,59,60,61,62,63,64]. Compared with other plants, leafy vegetables generally show higher detection ratio and concentrations of PPCPs [60,64] and therefore deserve more attention in their food safety risk. Although there are no official documents to explicitly clarify the MRLs of PPCPs in plant-derived foods, we can still deduce their safety thresholds from their corresponding MRLs in animal-derived foods [1,2,3,4]. Relative to the colossal number of analytical methods for PPCPs in animal-derived foods [65,66,67,68,69], the methods for PPCPs detection in plant matrices are in short supply. To better cope with the complicated PPCPs contamination in plants, the top priority is to develop a high-throughput screening method that can accurately, rapidly and comprehensively determine which PPCPs exist in the foods. With this consideration, we developed this novel metabolomics-based analytical method to achieve non-targeted screening of PPCPs in plant-derived foods.

## 4. Conclusions

The newly developed metabolomics analytical method was successfully applicable to non-targeted screening of 50 PPCPs residues in lettuce and maize matrices. We intentionally designed three concentration groups of PPCPs (20, 50 and 100 ng mL^−1^) to simulate the experimental and control groups adopted in the traditional metabolomics analytical procedures to search for marker compounds on behalf of 50 PPCPs. The process to perform metabolomics analysis has less artificial interference, a more concise workflow and higher screening efficiency. It is worth mentioning that this is the first implemented analytical strategy of metabolomics for non-targeted screening of PPCPs in plant-derived foods through seeking marker compounds. Due to the lack of binding legal documents on MRLs of PPCPs in plant matrices, together with constant development and application of new PPCPs in animal husbandry, it is urgent to compile legal rules to control MRLs of PPCPs in plant-derived foods, otherwise it may evolve as a serious food safety issue. To date, plant uptake from PPCP-contaminated soil is a known source of PPCP residues in plant-derived foods. It is not yet clear whether other ways can also induce the accumulation of PPCPs in the foods, potentially increasing the complexity of PPCPs contamination. Even worse, this increases the exposure risk of PPCPs to human health via the food chain. Therefore, we advocate that early attention to this issue would help defuse the potential crisis.

## Figures and Tables

**Figure 1 molecules-27-04711-f001:**
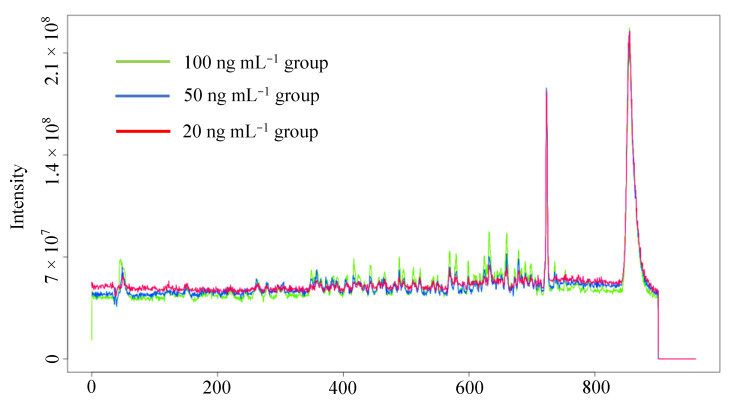
Total ion chromatograms (0~900 s) of spiked lettuce sample groups on the W4M platform.

**Figure 2 molecules-27-04711-f002:**
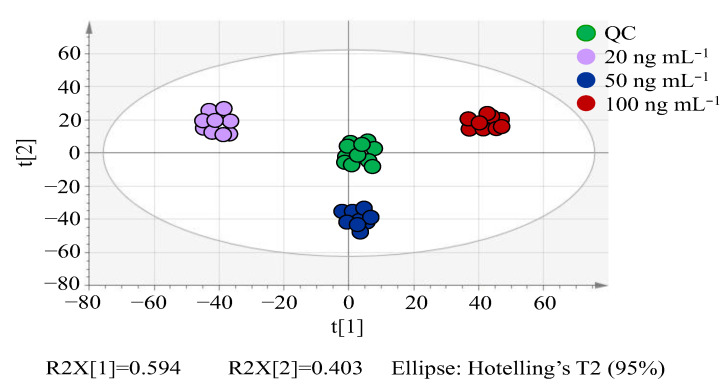
PCA score plot of spiked lettuce sample groups.

**Figure 3 molecules-27-04711-f003:**
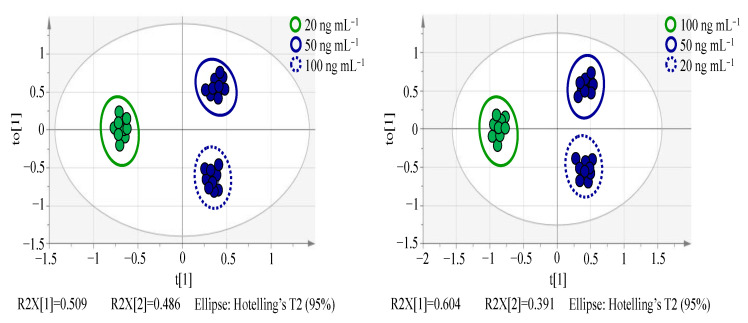
OPLS-DA score plots of spiked lettuce sample groups.

**Figure 4 molecules-27-04711-f004:**
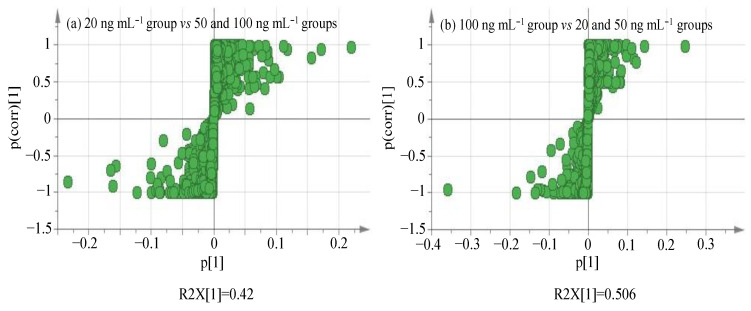
S-plot plots of spiked lettuce sample groups.

**Figure 5 molecules-27-04711-f005:**
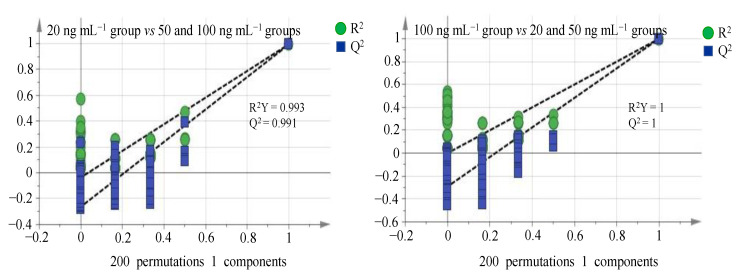
Permutation test plots of spiked lettuce sample groups.

**Table 1 molecules-27-04711-t001:** Basic information on the 50 PPCPs.

No.	Compound	CAS No.	Category	Adduct	Parent Ion (*m*/*z*)	Retention Time (min)
1	Clorprenaline	3811-25-4	Agonist	M + H	214.09932	7.03
2	Terbutaline	23031-25-6			226.14377	2.65
3	Tolobuterol	41570-61-0			228.11496	8.23
4	Cimbuterol	54239-39-3			234.16009	4.47
5	Propranolol	5051-22-9			260.16451	9.53
6	Sotalol	959-24-0			273.12674	2.47
7	Nadolol	42200-33-9			310.20128	6.47
8	5-Chloro-1-methyl-4-nitroimidazole	4897-25-0	Nitroimidazoles	M + H	162.00649	4.43
9	Ipronidazole	14885-29-1			170.09241	7.79
10	Metronidazole	443-48-1			172.07167	2.58
11	Metronidazole-hydroxy	4812-40-2			188.06658	1.69
12	Ronidazole	7681-76-7			201.06183	2.96
13	Thiabendazole	148-79-8			202.04334	5.83
14	Ornidazole	16773-42-5			220.04835	6.41
15	Tinidazole	19387-91-8			248.06996	4.96
16	Albendazole	54965-21-8			266.09577	10.48
17	Fenbendazole	43210-67-9			300.08012	11.06
18	Oxolinic acid	14698-29-4	Quinolones	M + H	262.07100	9.16
19	Cinoxacin	28657-80-9			263.06625	8.69
20	Norfloxacin	70458-96-7			320.14051	6.50
21	Enoxacin	74011-58-8			321.13575	6.35
22	Ciprofloxacin	85721-33-1			332.14052	6.73
23	Lomefloxacin	98079-51-7			352.14672	7.12
24	Danofloxacin	112398-08-0			358.15615	7.00
25	Enrofloxacin	93106-60-6			360.17183	7.00
26	Ofloxacin	82419-36-1			362.15106	6.39
27	Marbofloxacin	11550-35-1			363.14631	5.93
28	Sparfloxacin	110871-86-8			393.17327	8.34
29	Difloxacin	98106-17-3			400.14672	7.42
30	Trenbolone	10161-33-8	Steroid	M + H	271.16926	10.98
31	Boldenone	846-48-0			287.20056	11.02
32	Testosterone propionate	57-85-2			345.24242	12.30
33	Deflazacort	14484-47-0			442.22241	11.06
34	Tetracycline	60-54-8	Tetracyclines	M + H	445.13444	6.56
35	Oxytetracycline	79-57-2			461.13391	6.41
36	Chlorotetracycline	57-62-5			479.38028	7.88
37	Sulphacetamide	144-80-9	Sulfonamides	M + H	215.04849	1.99
38	Sulfapyridine	144-83-2			250.06447	4.68
39	Sulfadiazine	68-35-9			251.05972	3.37
40	Sulfathiazole	72-14-0			256.02089	4.32
41	Sulfamerazine	127-79-7			265.07537	5.11
42	Sulfamoxole	729-99-7			268.07504	6.10
43	Sulfamethizole	144-82-1			271.03179	6.11
44	Sulfabenzamide	127-71-9			277.06414	8.05
45	Sulfmethazine	57-68-1			279.09102	3.76
46	Sulfisomidine	515-64-0			279.09102	6.24
47	Sulfachloropyridazine	80-32-0			285.02075	6.80
48	Trimethoprim	738-70-5			291.14517	6.01
49	Sulfaquinoxaline	59-40-5			301.07537	9.26
50	Sulfanitran	122-16-7			336.06486	10.08

**Table 2 molecules-27-04711-t002:** Marker compounds screened in lettuce sample groups.

Var ID (Primary)	Marker Compounds	VIP Pred *^a^*	Coordinate in S-Plot *^b^*	Mass Error (ppm) *^c^*	LOD (µg kg^−1^)
M214T418	Clorprenaline	3.300/3.057	(−0.077, −0.911)/(0.057, 0.934)	−0.225	1.3
M226T148	Terbutaline	2.297/2.467	(−0.049, −0.969)/(0.049, 0.945)	−1.751	2.0
M228T491	Tolobuterol	4.045/3.311	(−0.098, −0.922)/(0.063, 0.929)	−0.294	0.6
M234T261	Cimbuterol	3.226/4.071	(−0.075, −0.934)/(0.082, 0.965)	−2.061	1.1
M260T571	Propranolol	3.710/4.505	(−0.089, −0.927)/(0.099, 0.945)	0.617	0.5
M273T138	Sotalol	1.513/1.453	(−0.025, −0.923)/(0.025, 0.916)	0.563	0.7
M310T388	Nadolol	4.165/2.141	(−0.101, −0.981)/(0.038, 0.933)	−2.351	0.7
M162T266	5-Chloro-1-methyl-4-nitroimidazole	1.854/1.675	(−0.038, −0.971)/(0.032, 0.948)	−0.394	0.6
M170T467	Ipronidazole	4.537/4.102	(−0.111, −0.921)/(0.087, 0.927)	2.920	0.4
M172T152	Metronidazole	2.838/2.675	(−0.064, −0.928)/(0.052, 0.967)	2.916	1.7
M188T101	Metronidazole-hydroxy	2.178/2.756	(−0.046, −0.964)/(0.053, 0.956)	−0.353	1.7
M201T179	Ronidazole	1.957/1.673	(−0.040, −0.961)/(0.032, 0.907)	−1.134	2.0
M202T351	Thiabendazole	3.879/4.056	(−0.093, −0.946)/(0.078, 0.986)	−1.654	0.9
M220T383	Ornidazole	3.627/1.928	(−0.087, −0.932)/(0.035, 0.914)	−1.476	1.4
M248T298	Tinidazole	2.977/3.588	(−0.069, −0.938)/(0.070, 0.952)	−2.146	0.4
M266T625	Albendazole	4.193/3.237	(−0.102, −0.918)/(0.060, 0.911)	0.143	1.7
M300T661	Fenbendazole	3.050/2.891	(−0.071, −0.946)/(0.054, 0.961)	−0.272	1.1
M262T553	Oxolinic acid	2.091/2.258	(−0.044, −0.932)/(0.040, 0.955)	4.435	0.5
M263T524	Cinoxacin	2.481/2.115	(−0.055, −0.986)/(0.037, 0.973)	2.106	0.7
M320T375	Norfloxacin	2.684/2.113	(−0.059, −0.966)/(0.037, 0.949)	0.600	0.6
M321T381	Enoxacin	2.479/3.480	(−0.055, −0.937)/(0.067, 0.924)	1.102	0.7
M332T406	Ciprofloxacin	3.110/3.589	(−0.072, −0.934)/(0.070, 0.948)	4.829	1.0
M352T427	Lomefloxacin	3.052/2.676	(−0.071, −0.928)/(0.052, 0.933)	1.175	1.6
M358T420	Danofloxacin	2.842/2.469	(−0.065, −0.929)/(0.049, 0.937)	2.019	1.8
M360T422	Enrofloxacin	2.836/2.397	(−0.064, −0.957)/(0.045, 0.959)	0.770	1.9
M362T386	Ofloxacin	1.553/1.672	(−0.028, −0.925)/(0.031, 0.966)	0.908	0.8
M363T386	Marbofloxacin	1.617/1.678	(−0.030, −0.923)/(0.032, 0.929)	−4.510	0.6
M393T500	Sparfloxacin	1.827/2.946	(−0.036, −0.911)/(0.056, 0.945)	−3.455	0.8
M400T446	Difloxacin	1.535/2.317	(−0.028, −0.937)/(0.042, 0.949)	−2.715	0.8
M271T659	Trenbolone	3.077/1.586	(−0.071, −0.947)/(0.029, 0.945)	1.869	0.7
M287T661	Boldenone	2.071/2.398	(−0.043, −0.944)/(0.045, 0.936)	−0.921	0.8
M345T738	Testosterone propionate	2.698/1.676	(−0.061, −0.928)/(0.032, 0.927)	0.122	0.8
M442T663	Deflazacort	1.535/1.545	(−0.026, −0.977)/(0.027, 0.919)	−4.677	1.0
M445T393	Tetracycline	4.047/4.059	(−0.098, −0.911)/(0.079, 0.941)	−0.539	1.0
M461T385	Oxytetracycline	2.708/2.677	(−0.061, −0.928)/(0.052, 0.934)	1.062	0.5
M479T473	Chlorotetracycline	2.180/2.392	(−0.046, −0.945)/(0.044, 0.928)	1.502	0.5
M215T120	Sulphacetamide	1.615/1.468	(−0.030, −0.915)/(0.026, 0.933)	4.277	0.7
M250T278	Sulfapyridine	3.108/4.087	(−0.072, −0.956)/(0.085, 0.927)	−0.686	0.7
M251T200	Sulfadiazine	2.075/1.893	(−0.043, −0.912)/(0.034, 0.921)	0.422	0.7
M256T260	Sulfathiazole	3.880/2.893	(−0.093, −0.945)/(0.054, 0.948)	0.280	0.9
M265T304	Sulfamerazine	2.415/3.483	(−0.053, −0.987)/(0.067, 0.987)	−2.799	0.4
M268T345	Sulfamoxole	3.301/3.312	(−0.077, −0.922)/(0.063, 0.909)	0.002	1.1
M271T366	Sulfamethizole	2.704/1.412	(−0.061, −0.956)/(0.023, 0.919)	−2.372	1.8
M277T482	Sulfabenzamide	1.883/2.417	(−0.038, −0.944)/(0.046, 0.951)	2.561	0.6
M279T222	Sulfmethazine	4.119/3.117	(−0.100, −0.927)/(0.058, 0.978)	−4.681	1.3
M279T374	Sulfisomidine	2.711/3.661	(−0.061, −0.946)/(0.072, 0.923)	−3.077	0.5
M285T407	Sulfachloropyridazine	1.639/1.569	(−0.031, −0.982)/(0.028, 0.958)	−2.350	0.5
M291T360	Trimethoprim	1.952/1.585	(−0.040, −0.945)/(0.029, 0.928)	0.465	0.6
M301T557	Sulfaquinoxaline	1.829/2.115	(−0.036, −0.934)/(0.037, 0.922)	−0.561	0.4
M336T672	Sulfanitran	1.535/1.568	(−0.027, −0.928)/(0.028, 0.927)	2.996	1.6

Note: *^a^* two VIP values from 100 and 20 ng mL^−1^ groups, respectively; *^b^* two-group coordinate values from 100 and 20 ng mL^−1^ groups, respectively; *^c^* Mass error (ppm) = (extracted molecular weight from W4M platform—extracted molecular weight from LC-MS/MS) × 10^6^/extracted molecular weight from LC-MS/MS.

## Data Availability

The data presented in this study are available in this article and Appendix A.

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
