# Peer review of "Metabolomics Approach on Non-Targeted Screening of 50 PPCPs in Lettuce and Maize"

_molecules, 2022, doi:10.3390/molecules27154711_

Round 1

Reviewer 1 Report

See the attached pdf for comments.

Author Response

Reviewer 1

These are three groups differenciated on concentrations. The meaning of the phrase ‘two camps’ is not clear here. A clarification by the authors are needed here.

Response: Thanks for your comment and suggestion! The suggestion has been adopted. We have revised the relevant expression. Please see Page 7 of 15, Lines 240 ~ 242 in the main text in the revised manuscript.

Reviewer 2 Report

some comments  

- To validate the results, the phytochemistry part is required. 

- The utility of Univariate Analysis not clear.

- Where is the discussion part? May be added and will be justify each activity.

- The experimental part will be reduced. It seems to me that the plagiarism is too higher.

Some references (38,55,68, etc…) will be updated

Author Response

Reviewer 2

some comments  

- To validate the results, the phytochemistry part is required. 

Response: Thanks for your comment! The main goal of this study is to develop a non-targeted screening method of various PPCPs in plant-derived foods based on metabolomics analytical strategy. Herein, the lettuce and maize were deemed just two plant matrices accommodating PPCPs, rather than main targets to investigate how the internal chemical reactions in the plants affect the screening of PPCPs. Phytochemistry may affect the screening results, but it is not the concern of our study. Moreover, the current analysis is able to explain the validation results. Therefore, we consider not to supplement phytochemistry analysis.

Reviewer 2

- The utility of Univariate Analysis not clear.

Response: Thanks for your comment! We have supplemented the utility of univariate analysis. Please see Page 6 of 15, Lines 194 ~ 196 and Page 8 of 15, Lines 275 ~ 277 in the main text in the revised manuscript.

Reviewer 2

- Where is the discussion part? May be added and will be justify each activity.

Response: Thanks for your comment! We have combined the “Results” section with the “Discussion” section to better clarify how our metabolomics screening method is developed and applied. The “Results and Discussion” section in this study is to describe the research results, on the basis of which, we further investigate and analyze whether the non-targeted screening method is workable. The screening method is developed based on multivariate and univariate analysis, the reasonability of which cannot be proved by the results alone, therefore we analyze the results to some extent. The analysis is relatively concise, but it is direct and enough to clarify the development process, application domain, feasibility and practicability of our method. Therefore, we prefer to keep the original analysis and discussion instead of adding new contents.

Reviewer 2

- The experimental part will be reduced. It seems to me that the plagiarism is too higher.

Response: Thanks for your comment! Previous studies (e.g., (a) Ahmed M.B.M., et al. Distribution and accumulative pattern of tetracyclines and sulfonamides in edible vegetables of cucumber, tomato and lettuce. J. Agric. Food Chem. 2015, 63, 398-405; (b) Hu X., et al. Occurrence and source analysis of typical veterinary antibiotics in manure, soil, vegetables and  groundwater from organic vegetable bases, northern China. Environ. Pollut. 2010, 158, 2992-2998; (c) Jones-Lepp T.L., et al. Method development and application to determine potential plant uptake of antibiotics and other drugs in irrigated crop production systems. J. Agric. Food Chem. 2010, 58, 11568-11573; (d) Pan M., et al. Distribution of antibiotics in wastewater-irrigated soils and their accumulation in vegetable crops in the Pearl River Delta, Southern China. J. Agric. Food Chem. 2014, 62, 11062-11069; (e) Sallach J.B., et al. Development and comparison of four methods for the extraction of antibiotics from a vegetative matrix. Environ. Toxicol. Chem. 2015, 35, 889-897.) have proposed some experimental procedures and analytical methods based on high-performance liquid chromatography-tandem mass spectrometer (HPLC-MS/MS) for PPCPs detection in plant matrices. These studies mainly focused on the contamination of antibiotics, especially for tetracyclines, quinolones and sulfonamides. In our study, we focus on more kinds of PPCPs to be screened in a single injection of UHPLC-MS/MS by our developed metabolomics analytical approach. We design the experiments based on previously developed methods, which are described by the common sentences we are likely to be familiar with, but due to the differences of PPCPs to be screened, plant matrices, equipment models, etc., we have to optimize the previous methods and propose new methods specific for our study, in which we have made some adjustments for the experimental procedures, equipment settings, analytical methods and even the sentence expression. It is worth noting that it is the first time to apply metabolomics approach to non-targeted screening of PPCPs in plant-derived foods, therefore, all the experiments should serve the metabolomics analysis, which probably requires some experimental parts different with those of other studies. The experimental parts in our study reflect the similarity and difference with those in previous studies, and make a good comparison among them. More importantly, each experimental step is closely related to the final metabolomics screening results, therefore, we think the experimental parts are essential to be described in detail, that is why we consider not to delete some experimental contents.

Reviewer 2

Some references (38,55,68, etc…) will be updated

Response: Thanks for your comment! According to the retrieved results from Google Scholar, Reference 38 ”Gika H.G., et al. Liquid chromatography and ultra-performance liquid chromatography-mass spectrometry fingerprinting of human urine: sample stability under different handling and storage conditions for metabonomics studies. J. Chromatogr. A 2008, 1189, 314-322” and Reference 68 “Schneider M.J., et al. Evaluation of a multi-class, multi-residue liquid chromatography-tandem mass spectrometry method for analysis of 120 veterinary drugs in bovine kidney. Drug Test. Anal. 2012, 4, 91-102” have been cited for 202 and 56 times, respectively, indicating their work is of relatively high quality. Reference 55 “Environmental Protection Agency (EPA). Method 8061A Phthalate esters by gas chromatography with electron capture detection (GC/ECD). 1996” has been implemented as an EPA standard for 20 more years, and it is still valid, proving the practicability of the work is still very high. More importantly, these above-mentioned references are qualified to support our views, therefore we consider not to update them.

Round 2

Reviewer 2 Report

- delete the keyword: pharmaceutical and personal care products

- not clear the section 2.4. Sample Grouping and Naming line139-140.; please Rewrite